# The Impact of In-Classroom Non-Digital Game-Based Learning Activities on Students Transitioning to Higher Education

Chitra Balakrishna 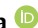

School of Comouting and Communications, The Open University, Milton Keynes MK7 6BJ, UK; chitra.balakrishna@open.ac.uk

**Abstract:** The initial phase of learning at a university has a bearing on students' long-term academic development and plays a crucial role in enabling them to successfully transition to higher education. While higher education institutes have long been struggling to address the challenge of student retention and student success, the new generation of learners (millennials and Generation Z) entering universities have brought in further complexity. This study explores the impact of in-classroom, *non-digital* game-based learning techniques on the academic performance, classroom engagement, and peer interaction among the first-year university students studying computing qualification. The study aimed to deduce how the overall enhanced learning experience of these students enables them to integrate into the new learning environment in the university, thereby helping them to successfully transition to higher education. Data for this study were taken from among the first-year computing students across two consecutive years of study (N = 251). The results corroborated the findings from previous studies and highlighted how academic performance, classroom engagement, and peer interaction considerably enhance students' academic integration. The study concludes with a discussion of the limitations and implications for practice and future research.

**Keywords:** non-digital; game-based learning activities; classroom engagement; GBL in the classroom; classroom interaction; learner experience; GBL for millennials; GBL for Gen Z; academic self-efficacy; academic integration; transition to higher education

## 1. Introduction

A common concern for higher education institutions (HEIs) across the globe is student retention and overall student success at the university. The factors that enable a successful transition from high school to university is a well-researched problem area, as demonstrated by a rapidly expanding literature [1–4]. The literature suggests that there is no single common factor that enables successful transition or causes student attrition. It is a combination of personal, social, cultural, and academic factors. The theoretical models of student retention and transition are strongly influenced by Tinto's student integration theory [5]. According to Tinto, students who integrate with the campus and university, both academically and socially, are more likely to stay and succeed in their programme of study. A large part of the students' connection to their campus is through engagement with their peer groups. These peer groups normally share a common ground such as studying similar courses, sharing residences, or are a part of a common extra-curricular group. Personal factors such as student's financial troubles, difficult personal circumstances, or dissatisfaction due to an uninformed choice of program or university [6] cannot be controlled by the HEIs. However, enabling the students to academically integrate with the university, i.e., by enabling them to engage in the classrooms, enhance their academic performance, facilitate peer interaction that may foster friendships as identified by Tinto, is something HEIs can control to support student learning and retention [7]. This amounts to creating a friendly and trusted learning community to overcome the fear and daunting experience university life thrusts on students transitioning to higher education.

One way the HEIs can foster learning communities among the transitioning students is by facilitating peer engagement and interaction in classrooms. A common challenge students in the first year face is to adapt to the new learning environment within HEIs, where there is the lack of continuous formal contact with their teachers. Some students find it daunting to learn independently and work on assignments beyond the classroom environment without academic supervision [8], and the lack of support from peers that most students develop over the period of their secondary education adds to the challenge. Researchers universally agree that the learning experience of first-year students has a direct impact on student retention rates and their overall success in studies [9]. It has, therefore, become imminent to review teaching practices within the first-year modules at the HEIs, to improve peer engagement and student participation to enable a smooth transition of students into higher education learning methods.

Effective teaching methods in higher education require an understanding and appreciation of the learners' needs, backgrounds, interests, and learning styles. There is an increasing number of young learners entering universities that are exposed to digital technologies all their lives, often referred to as "Generation Z" [10]. They are not particularly motivated by the non-interactive lecture-/presentation-based learning that is prevalent within the majority of higher education classroom settings. They demand more interaction with or without technology assistance. Consequently, there is an increasing pressure on educators to promote engagement in the classroom by being not just deliverers of content, but facilitators of the learning process [11]. This is often accomplished through active learning strategies such as the flipped classroom [12], technology integration, and the use of game-based learning methods [13].

According to Dominguez et al. [14] and Crocco et al. [15], education researchers have considered the use of games with great interest. The scholarship during the past two decades has primarily focused on the theory of game-based learning and why games are a powerful tool of instruction. Important work among them are from Prensky [13], Gee [16], Oblinger [17], and Squire et al. [18]. More recently, the work by Eltahir et al. [19], Jaaska et al. [20], and Martin-Hernandez et al. [21] positively illustrated the effectiveness of game-based learning methods within the higher education setting. The dominant argument by the scholars is that the "digital natives" [22] as defined by Prensky have developed radically different learning styles due to the extended exposure to digital and multimedia content including games and, therefore, would thrive in a learning environment that is multi-modal and feedback-rich, similar to digital games.

In this study, we explored the power of non-digital game-based learning methods implemented within the classroom setting among first-year higher education students, where gameplay techniques inform the design of classroom learning activities. The study in particular measures the impact of game-based learning activities on student engagement, peer interaction, their academic performance, and their classroom learning experiences, which, according to Tinto's student integration theory, will potentially contribute to the student's successful transition to higher education.

*Gamification and Game-Based Learning*

The terms "gamification" and "game-based learning" have often been used interchangeably within the context of teaching and learning. While there is no specific definition for the term "gamification", it is described as those features in interactive systems that aim to motivate and engage end-users through the use of game elements and mechanisms [23]. Gamification is broadly seen as the application of elements of gaming design and game mechanics in a non-game context [24], which manifest as reward points, leader boards, badges of recognition, etc. Alternatively, game-based learning (GBL) or serious games are methods where the game *is the* learning. The game-based learning activities are normally designed to achieve a specific learning objective or outcome [13]. In this paper, we investigated a pilot deployment of in-classroom game-based learning methods.

Despite the strong argument in favour of game-based learning methods, there is limited evidence on the effectiveness of these methods, in particular that which contributes to student's success at the university. According to Liu and Chen [25], most studies on game-based learning (GBL) have focused on digital games and their role in enhancing student motivation and participation as compared to conventional teaching methods. There are limited studies on non-digital game-based learning methods applied within the higher education classroom setting, in particular to foster peer interaction and student engagement with the aim to increase student retention. This study aimed to address this gap.

In this paper, we present the findings from a pilot study that was conducted as part of the common first-year module titled "Digital World". The specific unit of study, where the game-based learning activities were implemented, was a 5-week unit that taught introductory concepts of computer networking and security to first-year cohorts. The aim of the study was to assess the effectiveness of non-digital game-based learning activities in enhancing student engagement, improving academic performance, and fostering classroom peer interaction, leading to an enhanced learning experience. Based on Tinto's academic integration theory [26], student engagement, academic performance, and learner experience are key enabling factors for successful transitioning to higher education.

## 2. Motivation for the Study

As part of a common first-year program, "Digital World" was the first module students were enrolled in at the School of Computing. The module was split into three blocks, where each block was taught over a 3–5-week period. Each block addressed a specific sub-discipline within computing such as databases, programming, and computer networks and security. Computer networking and security were taught as the second block from Weeks 5–9 of the term. The motivation for this study primarily stemmed from the critical observations made during the end of module academic reflection following three consecutive presentations of the module.

As a part of the end of year reflection of academic practice, a variety of data such as the module attendance, module results, and end of module student evaluation were reviewed. The data analysis indicated that there was a consistent dip in student attendance and engagement starting at Week 5 of the module delivery. The average student performance in the block remained fairly stagnant across three years of delivery. Student feedback received as a part of the end of module evaluation indicated that the students were dissatisfied with the lecture-based delivery of the module and the module assignment. This was in contrast to their expectation of learning computer networking and security through hands-on and practical methods. The teacher-centric teaching practice employed in the module was ineffective among the students. This observation aligns with findings from studies relating to Generation Z [27], which indicates that Generation Z learners expect a teaching environment to simulate virtual worlds. They demand instant information, visual forms of learning, and replacing teacher-centric-instruction-based learning with interactive peer learning, challenges and problem solving, and creative activities.

The importance of the first year in higher education has long been recognised as being vital for student retention and their ultimate success in completing their study [28]. Studies indicate that, for students to successfully manage the transition into higher education, there is a need for their integration into the new learning environment. In this context, the work by Schaeper [29] provides evidence that a cognitively activating learning environment enhances academic integration considerably. The findings from these studies were a motivating factor in introducing interventions within classroom setting through the use of non-digital game-based learning activities in this study. The goal of these interventions was to create an engaging and interactive learning environment within the classrooms to foster student bonds, which potentially will extend beyond the classroom environment and enable the students to successfully transition into higher education. The study aimed to assess the effectiveness of the intervention in addressing the critical issues of student attendance and student performance.

Computer networking and information security as a subject typically requires a fine balance between delivering the theoretical concepts and hands-on practical components [30]. While concepts in this subject are best learned through hands-on lab activities, the learning will be incomplete without adequate understanding of the theoretical concepts to appreciate the practical tasks. Given that this module was a first-year introductory module, it was essential to meet the students' expectations while also not compromising on delivering the foundational concepts of the module. The primary motivation to use game-based learning methods within the module was to make the dry and boring theoretical topics interesting, engaging, and fun, while also fostering in-classroom interaction to enhance the overall student learning experience. The intent was to assess if the cleverly chosen gameplay mechanics would lend itself naturally to drive the theoretical concepts, while also keeping the students motivated to perform the practical tasks.

### 2.1. Pedagogical Motivator

Prensky [22] in his work defines the learners exposed to digital technologies all their lives as "digital natives". Prensky's work claims that these learners have acquired new cognitive capabilities and learning styles due to extended exposure to digital technologies and need to be motivated using new pedagogical methods. Dede [31] in his study indicated that these young learners have acquired new learning styles such as "fluency in multiple media, valuing each for the types of communication, activities, experiences, and expressions it empowers". Traditional teacher-centric practices that are driven by Objectivism [32], where the teacher transfers objective knowledge to the learner, are ineffective in engaging the new generation of learners. This new generation of learners is termed Generation Z, which is "hyperconnected" and intuitively familiar with computers and the Internet and consists of active problem solvers, independent learners, but also seeking social interaction and peer learning. There is an imminent need to address the evolving learning styles among the Generation Z learners.

Recent findings from the study conducted by Hazim et al. [33] indicated the effectiveness of game-based learning methods among the Generation Z learners. The majority of the Generation Z students perceived game-based learning as an effective method of instruction. They agreed that game-based learning motivates them to engage in learning, do better, and at the same time, encourages critical thinking and teamwork. All characteristics will enable students to successfully transition to higher education. The cohort in the "Digital World" module that are part of this study represent Generation Z, who were best-suited to investigate the effectiveness of game-based learning activities. Given that it was a common first-year module, the demography of learners within the module typically had a wide spectrum of learning abilities and varying levels of motivation. The game-based learning activities were aimed at addressing this challenge of heterogeneity in study abilities and motivation.

### 2.2. Learning Theories and Game-Based Learning

There are four main learning theories, and their representative principles are chronologically presented and identified as behaviourism, cognitivism, humanism, and constructivism [34]. As illustrated in Figure 1 [35], each learning theory has its own representative learning principles. In the context of this study, the constructivism approach [36], which views learning as a process of knowledge construction, with concept development and comprehensive understanding as the goals, lends itself naturally to a learner-centric practice. Constructivism advocates that effective learning occurs when the learners are actively involved in the construction of knowledge through interactive and experiential learning activities. Constructivism is instantiated within game-based learning. Constructivism focuses on the importance of the socio-cultural context in understanding what occurs in the world through social interaction and constructing knowledge. Studies [37] reveal strong empirical evidence that active involvement in the learning process is critical for improved learning experience and enhances the likelihood of the learner persisting and successfully

completing the qualification. The focus of the this study was to examine the enabling factors for first-year university students to actively engage in their learning, resulting in improved academic performance and retention.

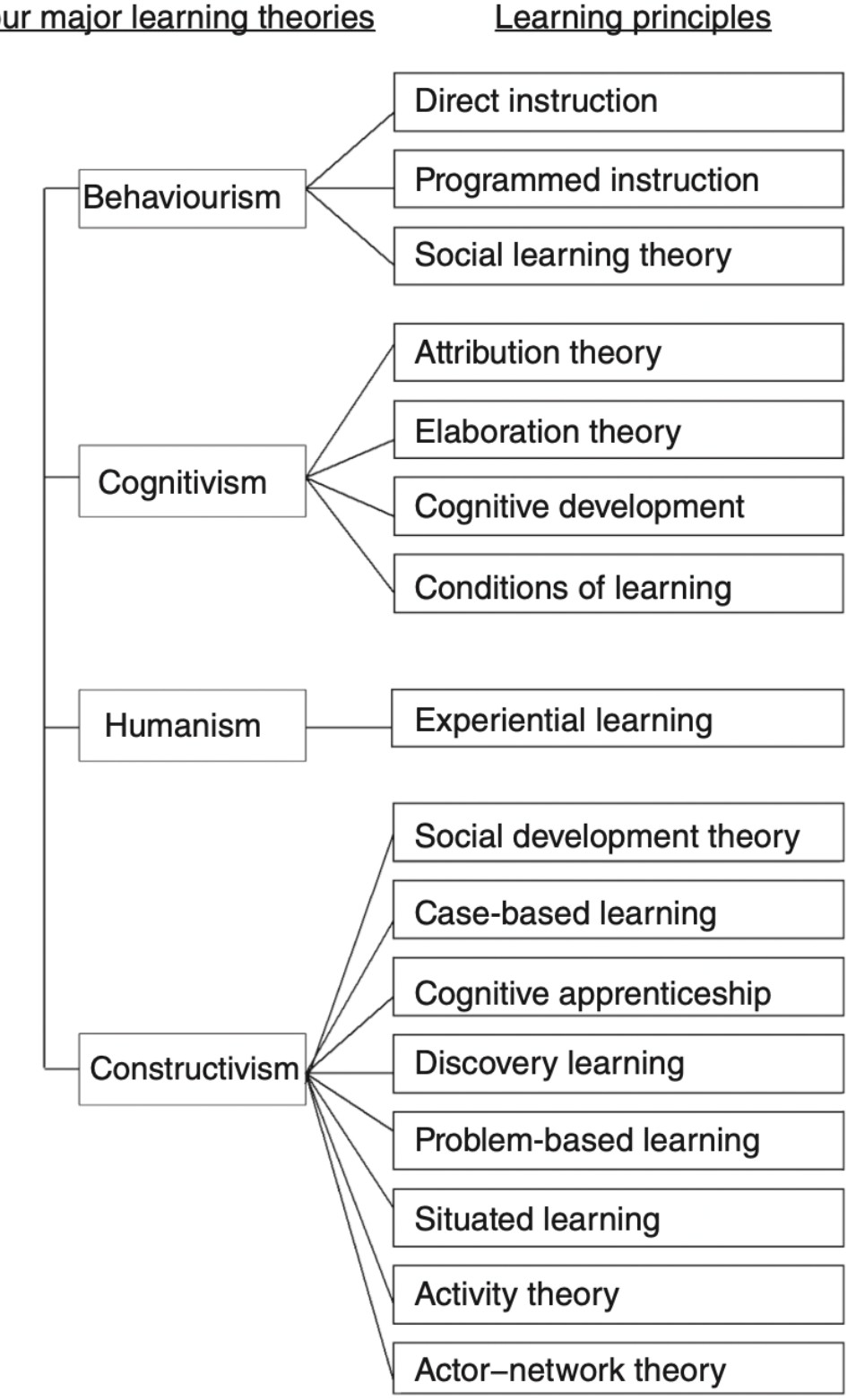

**Figure 1.** This figure illustrates the learning theories and associated learning principles.

In an effort to transform the current lecture-based delivery in the module into an active and evidence-driven delivery, classroom learning activities were designed based on game-based learning techniques. The activities were based on the core principles of constructive learning theory by allowing students to create their own knowledge by active interaction with the game-based learning activities. Reference [38] analysed several years' worth of student performance data, revealing that interactive learning activities relevant to the lectured material that are feedback driven improve student learning and classroom engagement. It is in this context that the pilot study was conducted.

*2.3. Research Questions*

The basis for this study was underpinned by Tinto's theory of academic integration [39], which indicates that enabling the academic integration of students results in their successful transition into higher education. The key objective of the research was to investigate the impact of in-classroom non-digital game-based learning activities on key enabling factors for academic integration of Generation Z students such as peer interaction, student engagement with the learning content, class attendance, and academic performance, which potentially could lead to student's learning commitment within the classroom, consequently enhancing student retention and their success in higher education. The study further made an informed recommendation to incorporate such gameplay elements in other modules that are largely technical in nature, where teachers face challenges in communicating the underlying theoretical concepts and in fostering classroom participation and student engagement. The following are the key research questions of the study:

- RQ1: What is the impact of in-classroom non-digital game-based learning activities on academic performance of first-year computing students transitioning to higher education?
- RQ2: What is the impact of in-classroom non-digital game-based learning activities on the peer interaction, engagement, and their participation in learning among the first-year computing students transitioning to higher education?
- RQ3: How do in-classroom non-digital game-based learning activities impact overall learning experience of first-year computing students transitioning to higher education?

**3. Literature Review**

Game players regularly exhibit persistence, risk-taking, attention to detail, and problem solving; all behaviours are ideally suited for effective learning [40]. Mark Prensky [22] presented a list of characteristics of gameplay that lends itself well to a learning scenario:

- Games are fun and give enjoyment to the players, thereby enabling passionate involvement, which is crucial for learning.
- Games have rules, goals, and are interactive, offering the structure, motivation, and engagement that is required for effective learning.
- Games are adaptive and have outcomes and feedback offering the flow and tangible outcome for learning.

The use of serious games concepts within education allow players to be wrong without negative consequences. There are fundamentally fewer risks when playing a game, and this allows players to try different strategies and new approaches by controlling the fear of failure [41]. This interactive trial and error learning method can lead to individualised learning; this can be in many contexts, for example taking different routes to a destination to find which one is the quickest or which one is the most-effective. Lastly, gamification enables students to engage in enjoyable experiences for the purpose of learning [14]. Connolly et al. [42] presented a systematic literature review on game-based learning and serious games focusing on positive outcomes. Their study stresses the need for more rigorous evidence of games' effectiveness and real impact. Extensive studies have been carried out on the use of game-based learning methods in school education and higher education alike [25,43]. These findings from these studies indicate a comparative improvement in

student engagement and student performance among students who were taught using a game-based learning strategy compared to the students taught using traditional methods. Some of these studies [43,44] examined the impact of game-based learning methods on educational outcomes and the performance of students and confirmed that game-based learning could promote students' academic achievement. Numerous studies have demonstrated a significant relationship between game-based learning and learners' motivation for learning [43,45,46].

The impact of educational gamification has been reported in studies, with significant results and positive feedback from both teachers and students. Ejsing et al. [47] concluded that a playful approach in higher education demonstrates the use of gamification, when balanced with the formal curriculum, allows the intrinsic motivation of the students towards deep learning while enhancing participation and engagement. The NMC Horizon Report (2014) [48] and similar reports indicate that educational gameplay has proven to nurture engagement in critical thinking, creative problem-solving, and teamwork—skills that lead to solutions for complex social and environmental dilemmas. This is particularly relevant within the context of this study where students go through similar social and environmental dilemmas while transitioning into higher education. Hence, a close look at the various game-based learning techniques and mechanics will help ensure the right approach is considered to achieve significant results in enhancing engagement amongst students.

*3.1. Game-Based Learning for Student Motivation and Engagement*

Motivation in this instance is described as the provision of an incentive to engage in the act of gaining knowledge [49]. Shu [50] defined engagement as a series of goal-oriented behaviours and reflections that represent a deep involvement in learning activities. Classroom tasks that earn points or marks to rate/reward a student's performance do not always translate to motivation, as most studies indicate. In fact, the introduction of such gamification methods within classrooms environments has become undesirable and is reflected in undesirable outcomes, such as disengagement, cheating, learned helplessness, and dropping out [51]. Therefore, it is critical to use the right game mechanics in the design of game-based learning activities employed within classrooms to trigger learner motivation. Fogg's Behaviour Model [52] states that game mechanics can be used to motivate and trigger desired behaviours among students. Although he provides a list of game-based learning elements explaining how they could be included in an e-learning course, there is no empirical research to prove the same. More work is required to demonstrate the evidence of impact and to trigger the large-scale implementation of game mechanics in pedagogy.

Engagement has been identified as a significant issue with existing training methods among adult learners [53]. Games are known for being engaging. Hamari et al. [54] found a positive correlation between challenges within game-based learning and the engagement of participants, which further correlates with perceived learning. A particularly important concept that relates to engagement is that of flow. Flow is a state of increased focus, immersion, and efficacy, brought about by a suitably engaging activity [55]. There have been numerous studies into flow and its impacts on engagement, learning, immersion, and enjoyment. Webster, Trevino, and Ryan [56] provided evidence suggesting that flow has a positive impact on learning and helps increase engagement with learning. More recently, the work by Rui Silva et al. [57] is relevant in this context. The study examined whether game-based learning resources increase the performance of undergraduate students studying accounting and marketing. Findings from the study indicated that there was a correlation between specific game characteristics and flow. The study concluded that, by introducing games into the curriculum, students' motivation and interest increased and demonstrated that games can be an effective way for students to learn.

Challenge is another significant factor in the induction of flow, as Jin [58] determined that challenge contributes towards flow, but only when the challenge is matched to the skill level. Another relevant study is by Kotob and Ibrahim [44], which aimed to verify the

impact of applying game-based learning to student motivation and academic performance in learning Arabic, confirming that game-based learning improved student motivation, engagement, and academic performance.

There are contradicting results on game-based learning that has resulted in disengagement and demotivation. Hanus and Fox [59] found that students in a gamified course showed less motivation, satisfaction, and engagement over time when compared with those who received traditional teaching techniques. Their study, however, was influenced by the competitiveness of students who engaged in the gamified course, which was perceived as either constructive or destructive competition. The competition factor was detrimental to some students, and they lost interest over time, whereas some students found that competition influenced their engagement and participation in the gamified course.

Domiguez et al. [14] in their study showed that students who completed the gamified experience obtained better scores on practical assignments, but performed poorly on written assignments. The students were also found to have an initial higher motivation in participation, which decreased with time. The assessment, however, was found to play a huge role in the students' performance compared to their actual learning experience or whether the gamified course facilitated deep learning. Some of the students also found the gamification plugin for the Blackboard e-learning platform difficult to use, and their motivation declined over time.

Hamari et al. [60] used gamification to support learning and development amongst students in a class. Although the students completed the gamified courses successfully and generated successful achievements, it was observed that students' motivation and participation declined over time.

The impact of educational gamification has been considered in related studies with significant results and positive feedback from both teachers and students. Ejsing-Duun et al. [47] concluded in their study that the playful approach in higher education demonstrates that the use of gamification if balanced with the formal curriculum allows the intrinsic motivation of students towards deep learning, while enhancing participation and engagement. Reports such as the NMC Horizon Report [61] indicate that educational gameplay has proven to nurture engagement in critical thinking, creative problem-solving, and teamwork—skills that lead to solutions for complex social and environmental dilemmas. Hence, a close look at the various gamification techniques and game mechanics will help ensure the right approach is considered to achieve significant results in enhancing learning and engagement amongst students.

The mechanics of collaboration between students in groups turns learning into an enjoyable activity and promotes social interaction while learning. In order to make the classroom familiar and more relaxing, Farzon and Brusilovsky [62] suggested that the use of social elements of games will simulate engagement within the classroom environment and also motivate students through peer interaction [63]. The aim of game mechanics is to use social dynamics to motivate students through influences such as competition with others and validation of peers. The basic desire of humans to achieve social acceptance, status, and recognition could be borrowed by the collaborative games, which could motivate students to achieve more by participation and engagement. The collaborative games by design aim to use learning to shape students' behaviour towards being social and collaborative by way of commenting, sharing, and peer marking [64].

### 3.2. The Gap

The majority of the literature in the field of game-based learning within the context of higher education study and the computer science discipline is based on digital game-based learning methods, while there were a few non-digital game-based learning (NDGBL) methods in subject areas other than computing. Fang et al. [65] in comparing digital game-based learning (DGBL) and NDGBL found significant differences between the two. The study indicated students felt more familiar, sympathetic, and satisfied when playing traditional non-digital games. Rahutami et al. [66] further indicated that NDGBL influenced

the learners in a more holistic way due to the direct contact involved such as visual, speech, and body in NDGBL, unlike in DGBL, where contact is limited to speech/audio. Hence, the study inferred that NDGBL resulted in better outcomes with regard to societal factors, critical thinking, cooperation, communication, and respect for opponents. In general, the findings from past studies found that NDGBL improved performance by providing a more enjoyable and active learning environment, which is particularly relevant in the context of this study to help students successfully transition into the higher education learning environment.

Apart from creating an engaging learning environment, a well-planned NDGBL approach can improve interaction skills, teamwork, investigative skills, information evaluation, and decision-making [67]. There have been several studies conducted in the past that examined the role of the NDGBL approach in various subjects such as English grammar [68] biology [69], chemistry [70], and accounting [71], and the benefits of NDGBL have been confirmed in those studies. However, there are very few studies that have examined the impact of non-digital classroom-based game-based learning methods on student engagement and learning experience among computing students or in the context computing-related courses [72,73]. It is this gap that is addressed in this study, which aimed to measure the impact of non-digital game-based learning methods on student engagement, peer interaction, and learning experience within the classroom setting among students studying first-year computing qualification. The scope of this work is limited to the classroom and does not extend to the VLE.

This research study was motivated by previous studies from an empirical point of view. The wide array of studies on the impact of game-based learning on student motivation, engagement, and overall learning experience indicate that positive outcomes are dependant on the demography of learners, the specific game mechanics chosen and applied, as well as on the clear purpose for using game-based methods within learning contexts.

## 4. Game Design

For each of the four teaching weeks, in-classroom game-based learning activities were designed. The core game mechanics chosen for the activities was based on the topic being taught and the specific purpose the gameplay elements were meant to serve. The game characteristics chosen for the learning activities for each week is described as follows:

- Week 1—collaboration-based gameplay: The aim of the Week 1 activity was to set the scene for the unit of study, making students recognise the need for data networking with a simple real-world problem demonstrating how the world would be without computer networks. As a first topic of the module, it was important to have an ice-breaker to get the learners to engage with the learning and with their peers. The "collaboration" game element enables the participants of the game to work together as a team to achieve the goals defined by the game, sharing the payoffs and outcomes [74]. The learning activity for the week involved viewing an animation video of a real-world scenario. Students working in groups of three had to identify the list of challenges from the demonstrated real-world scenario. They had to discuss, debate, and agree among themselves which of the listed problems they considered to be the top five problems. The learning activity was designed to tap into the potential of collaboration game mechanics, specifically to act as an ice-breaker during Week 1 of the module and for the students within the classroom to develop soft skills such as communication and teamwork, which collaborative games facilitate [75]. In order to make the activity a little more fun, the element of competition was added with the team that had the maximum number of problems matching the other teams being declared the winner. The competition element, in particular when implemented as a team activity, had a positive impact on peer interaction and engagement.
- Week 2—role-play game (RPG): RPGs offer immersion by giving the players the opportunity to take on characters, which helps them form a sense of belonging with other players, as well as with the content [76]. The Week 2 lecture material delivered

the theoretical concepts of computer networking, which the learners perceived as dry and boring during the previous presentation, as indicated by the end of module evaluation. The RPG mechanics was specifically chosen to address the challenge with learner attention and engagement in the classroom. The lecture-based teaching content was transformed into a fun, interactive, role-play-based learning activity in Week 2. Students worked in teams, where they assumed the role of a workstation, and each team formed human chains of various network topologies and learned about the relative strengths and weaknesses of each type of network topology. The element of competition was introduced, where each team solved a given real-world problem and worked out the cost for the proposed solution, using the knowledge gained from the role-playing exercise. The team with the most-economical solution won the competition. While the RPG element made the learning fun and engaging, the competition element motivated the students to apply the skills learned to solve a real world problem.

- Week 3—challenge-based gameplay: The core principle of challenge-based game design is centred on overcoming challenges, progressing and earning rewards, and feeling competent. According to the self-determination theory, the sense of achievement derived through challenge-based games is associated with an intrinsic motivation for students and maximises their knowledge acquisition [77]. In Week 3, the students were taught about the various network tools and techniques (ping, traceroute, whois, nslookup, etc.) through a lecture presentation. Following the lecture, students were made to complete a challenge-based learning activity working in pairs. Each partner was required to challenge the other by providing a name of a random website. The challenge was to obtain as much information about the given website as possible using the investigative networking tools taught in the lecture material. The student that gathered the maximum network information won the challenge. The challenge elements made the students probe each of the tools learned and apply them appropriately, while also enhancing the overall engagement with the content and enabling collaborative peer learning.
- Construction-based gameplay: Week 4 introduced Cisco's Packet Tracer simulation software, where the students learned to build simple networks and test the network operation on the software. Although there was no explicit gameplay element applied in this task, the Packet Tracer as the software emulated a construction toy such as Legos. Studies have indicated that construction-based gameplay enhances learning outcomes significantly within classrooms [78]. The Packet Tracer-based learning activity enabled the students to learn about the basics of network design by constructing networks in a trial and error mode with Packet Tracer as a playground. This also offered the students the hands-on practical mode of learning.

## 5. Research Philosophy

A research philosophy is a belief about the way in which data about a phenomenon should be gathered, analysed, and used. This study agrees with [79] that the consideration of the philosophical underpinning can be vital for shaping the research design and for explaining the approaches taken in order to support the credibility of research outcomes. It deals with the interaction of technology and people and will focus on statistical methods to make generalisations on measurements of the various game-based learning factors that impact the student learning experiences. Positivistic research emphasises quantification in the collection and analysis of data, has a deductive relationship between theory and research, and has an objectivist conception of reality. Furthermore, quantitative methods generally use standardised measures with predetermined response categories [80].

The mixed methods approach was used in this study, which involves the use of both quantitative and qualitative approaches. The explanatory mixed method research employed in this study started with collecting and analysing quantitative data. This was followed by a qualitative data gathering to learn and explain more about the context behind the figures

through surveys, focus groups, etc., as illustrated in Figure 2). Explanatory mixed method research integrates all of the findings for a broader and deeper understanding of the study participants. In this context, the study intended to better understand the perception of the participants in the experimental group to assess the impact of game-based learning methods on the various quantitative data gathered such as module attendance and the module score. This research design is especially useful when there is a need to explain and interpret quantitative findings. Quantitative research is flexible and allows one to collect data on different phenomena that is usually not in qualitative form such as attitudes and opinions, but can be converted to quantitative data, which makes it easier to analyse the data statistically. Qualitative research uses a systematic and rigorous approach that aims to answer questions concerned with what something is like, in our context student experience, and what students think or feel about something that they were exposed to, and it may address why something has happened as it has. Qualitative data often take the form of words or text and can include images. In our study, we captured qualitative data through a compulsory end of module survey and an optional survey for the experimental group. Qualitative research covers a very broad range of philosophical underpinnings and methodological approaches [81]. The positivist believes that the knowledge is universal and absolute, and based on their knowledge claim, positivists adopt the quantitative method to state reality in the world, whereas constructivists espouse a qualitative method to construct the meaning of the phenomena under investigation.

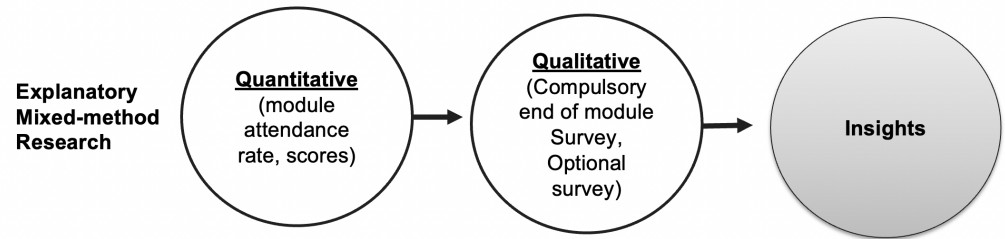

**Figure 2.** This figure illustrates the research method employed in this study.

### 5.1. Participants

The study participants consisted of 251 first-year university students studying computing. There was no specific sampling method employed in the study, as the study was conducted as part of the compulsory networking block spread across five weeks of study in the "Digital World" module. In this study, we used two groups of students (experimental group = 130 students from the current year of study (2018), who were taught using the in-classroom game-based learning activities; control group = 121 students from the previous year's cohort (2017), who were taught through traditional lecture presentations). Parameters considered for comparison were the average scores achieved by the experimental group and the control group, particularly the score achieved in the group element of the formative assessment to measure the learners' performance. The average attendance rate across the five weeks of the study unit between the two cohorts and the compulsory end of module evaluation survey responses received by the two sets of cohorts was considered to compare the in-classroom learner engagement and learner experience. The experimental group further participated in a voluntary survey, in order to measure the impact of the in-classroom game-based learning activities on their classroom interaction, engagement, and learning experience. Only 64 students out of the 130 students in the experimental group completed the optional survey questionnaire, which represents 25.49% of the total population.

Table 1 shows the demography characteristics of the participants.

**Table 1.** Participants' demography.

| Group | Demographic Parameter | Number (Percentage) |
|---|---|---|
| Experimental | First-year computing student cohort from current academic year | 130 (51.78%) |
| Control | First-year computing student cohort from the previous academic year | 121 (48.22%) |
| Gender | Male/female | 70 (27.88%)/181 (71.22%) |

*5.2. Research Design*

The quasi-empirical research design was used due to its appropriateness for the aims of the study. In this study, the researcher used two groups of students (experimental group = 130 students; control group = 121 students) to measure the academic performance and engagement of the cohort in a formative assignment at the end of the study period. The formative assignment consisted of completing the portfolio of all the learning activities completed in the classroom and had a group component with 40% weighting, while the remaining 60% of the assessment was the individual component. The formative assessment completed by the experimental group was based on the in-classroom game-based learning activities, while the formative assessment for the control group was based on traditional learning activities taught using lectures/presentations. The study in particular compared the performance in the group component of the formative assignment across the two study groups. The study measured two additional parameters to measure learner engagement within the classroom and with the learning content, namely the average weekly classroom attendance and the response to the compulsory end of module review questions relating to the module content. In addition, participants from the experimental group further completed a survey indicating their experience of the game-based learning activities in-classroom. Figure 3 below illustrates the research design employed in this study.

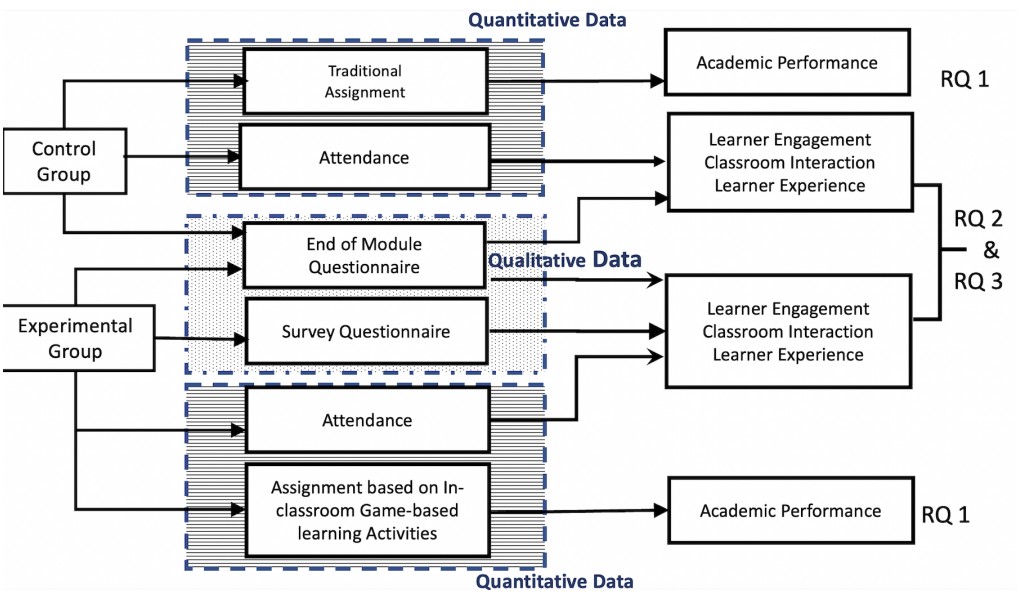

**Figure 3.** This figure illustrates the research design of the study.

*5.3. Equivalence (Validity) of Control and Experimental Groups*

In order to examine the equivalence of the participants in the study, two important data were captured for both participant groups. The two sets of data were then compared to indicate the equivalence between the two groups of participants and are illustrated in Table 2:

- Average academic score across three other first-year modules for the control group and the experimental group was captured according the academic bands and the number of non-submissions. As illustrated in Table 2, the difference in the average percentage scores between the control group and the experimental group was in the range (0.39–0.76%).
- Average classroom attendance across the three other first-year modules was captured, and the difference in average percentage attendance between the control group and the experimental group was in the range of (1–5%), which was considered to be a small difference in the context of this study.

**Table 2.** Control vs. experimental group data for equivalence.

| Demographic Parameter | Control Group | Experimental Group |
|---|---|---|
| Average score across other first-year modules (90 credits) | Non-submissions 12 (9.91%) fail— 15 (12.39%)/2:2—52 (42.14%)/2:1—27 (22.31%)/first—15 (12.39%) | Non-submission 11 (9.23%)/fail—17 (13.01%)/2:2—54 (41.53%)/2:1—30 (23.07%)/First—17 (13.01%) |
| Average attendance across other first-year modules | Module 1 (79%)/Module 2 (69%)/Module 3 (71%) | Module 1 (74%)/Module 2 (70%)/Module 3 (72%) |

This confirmed that there was no significant difference between the two study groups (experimental group and control group) and that the two groups were equivalent before the experimental method was applied or the game-based learning intervention was applied.

*5.4. Survey Instrument*

The data for the research were drawn from the first-year students using the following two survey instruments:

5.4.1. End of Module Questionnaire

The end of module (EoM) questionnaire was compulsory to complete for both the control group and the experimental group of participants. Since the module is taught as a common module for all the first-year students within the Computing Department, the data collected are highly representative of the whole population of all subjects who participated. The EoM questionnaire is broadly divided into four sections, three of which are standard with standard questions across all modules (learning materials, assessment, learning support), while the fourth section on learning experience is specific to this module. The standard sections of the EoM survey were authored by the faculty, whereas the module chair authored the questions relating to the learning experience. Only five out of the fifteen questions on the EoM questionnaire are relevant and were considered in this study:

- Q1. The lecture materials and classroom learning activities provided you with sufficient support to complete the formative assessments
- Q2. How difficult (overall) did you find this module?
- Q3. How actively did you participate in the in-classroom activities?
- Q4. What did you like about this module?
- Q5. What suggestions can you offer that would help make your learning experience better in this module?

5.4.2. Optional Survey

The optional survey was designed to measure the impact of the in-classroom game-based learning activities on students' learning experience, classroom interaction, and engagement from among the experimental group of participants. In order to assess participants' perception of the experimental method adopted, a Likert-scale-based questionnaire was administered. This explored the students' perceptions on the following aspects across all four game-based learning activities implemented in the study:

- Q1. The game-based learning activities made the module interesting.
- Q2. The game-based learning activities helped you understand the concepts better.
- Q3. The game-based learning activities helped you perform better on the the assignment.
- Q4. The collaborative and group game-based learning activities encouraged you to better interact with your peers in the classroom.
- Q5. The classroom interaction fostered by the game-based learning activities extended to out-of-class engagement with peers.

The Likert levels adopted by the researchers of this study were: strongly agree (5), agree (4), neither agree, nor disagree (3), disagree (2), and strongly disagree (1). Random sampling was used to select the sample size for the optional survey. The survey participation was partial; therefore, random sampling enabled reaching clearer conclusions due to its unbiased selection and the fact that it is highly representative of the population. The data were collected through a survey performed via self-administered online questionnaires. An invitation to participate in the survey was sent out to the students by email with clear instructions that the participation was entirely optional and would have no implication on the marking of the module. The questionnaire's reliability and validity were considered using the omega index, which was calculated to be 0.73, showing a high reliability and level of internal consistency. The data collected from the questionnaires were quantitatively analysed by calculating the frequency, mean, and standard deviation.

## 6. Results and Findings

### 6.1. Findings Related to Research Question 1

RQ1: What is the impact of in-classroom non-digital game-based learning activities on academic performance of first-year computing students transitioning to higher education?

Based on the research design illustrated in Figure 3, the data that were relevant to RQ1 were the average academic score in the module assignment including the split score in the group component of the assignment across the two participant groups. In addition, Q1 from the end of module questionnaire that was completed by both participant groups and Q3 from the optional survey questionnaire that only the experimental group completed were considered in this context. The findings are summarised in Table 3. The average score in the assignment for the experimental group was 74%, while the same group of participants scored an average of 65% across the other three modules in the first year. This score was 7% higher than the control group, whose average score in the module assignment was 67%, which is similar to the group's average score across the other three modules in the first year at 65%. This indicates that the spike in the average academic score of the experimental group in this module was due to the in-class game-based learning activities that the group was exposed to. This was further substantiated by the 4.54 (strongly agree) Likert response for Q1 from the end of module questionnaire (Q1: The lecture materials and classroom learning activities provided you with sufficient support to complete the formative assessments) among the experimental group, while the control group participants' score for Q1 was only 3.21 (weakly agree). Additionally, the response for Q3 from the optional survey questionnaire (Q3: The in-classroom game-based learning activities helped you perform better on the assignment) was 4.36 (strongly agree), supporting the inference that the in-classroom game-based learning activities had a positive impact on the overall academic performance of the experimental participant group. The first part of the survey queried the prior knowledge and background of the participants, and the responses indicated that the majority of respondents (38.2%) had moderate knowledge and 29.4% had close to no prior knowledge of computer networking before the module's attendance. Only a minimal population, i.e., 2.9% of the respondents, had expert knowledge of the module. This indicates that in-classroom game-based learning activities may be a strong influencing factor on the student's academic performance within the study block. This finding is in alignment with another similar study by Hamari et al. [54], who reported students having experienced higher immersion and intrinsic motivation to learn through game-based learning methods. Since they were willing to spend more time learning, their level of concentration was

also higher as compared to the non-gamified group, which resulted in better academic results. Similar findings were also reported by studies performed by Sawyer et al. [82]. It may be inferred that the introduction of these non-digital game-based interventions helped transform a traditional instructor-focused learning environment that the control group was exposed to into a student-centric one. As indicated by Ahmed et al. [83], serious game experiences transform the classroom learning environment where learners have the motivation to achieve and increased commitment to learn. The average score in the group component for the experimental group participants was 86 compared to 71 for the control group participants, which indicates that the collaborative and interactive in-classroom game-based learning activities facilitated the participants in the experimental group to perform better in the group component of the assignment compared to the control group counterpart, who were not exposed to any classroom-based interactive learning activities. This finding aligns with the results from similar studies reporting how GBL stimulates the collaboration between students. Collaborative environments often support students in their activities for learning. Collaborating in problem-solving encourages reflection, since students communicate, argue, and give opinions, and this enhances the learning process, resulting in improved academic performance [84].

**Table 3.** Findings related to RQ1.

| Data | Experimental Group | Control Group |
|---|---|---|
| Average assignment score mean (Std dev) | 74 (11.41) | 67 (11.02) |
| Average group component score mean (Std dev) | 86 (8.12) | 71 (9.80) |
| Average score across three other modules (Std dev) | 65 (10.63) | 69 (10.42) |
| EoM Q1. The lecture materials and classroom learning activities provided you with sufficient support to complete the formative assessments | 4.54 (strongly agree) | 3.21 (agree) |
| Survey Q3. The in-classroom game-based learning activities helped you perform better in the assignment | 4.36 (strongly agree) | Data not collected |

## 6.2. Findings Related to Research Question 2

RQ2: What is the impact of in-classroom non-digital game-based learning activities on the peer interaction, engagement, and their participation in learning among the first-year computing students transitioning to higher education?

Classroom attendance is a good indicator of learners' engagement and participation in learning. The data that were relevant for RQ2 were the average attendance score in this module and the average across the other three first-year modules. In addition, Q3 from the end of module questionnaire and Q1 and Q4 from the optional survey were considered in this context. Table 4 summarises the findings related to RQ2. The average percentage weekly attendance in the module was substantially higher for the experimental group (83%) compared to the control group (74%). The control group's average weekly attendance was similar to the average weekly attendance across three other first-year modules with a variance of ±5%. However, the average attendance for this module within the experimental group had a much higher variance compared to its average attendance across three other first-year modules (±13%). This is further substantiated by the high positive response (4.71—very actively) for Q3 on the end of module questionnaire (Q3: How actively did you participate in the classroom activities) among the experimental group compared (3.2—somewhat actively) with the control group. This indicates that the in-classroom game-based learning activities that were designed to be collaborative and interactive may have inherently influenced/motivated the students in the experimental group to attend the classes more regularly, whereas the control group had no collaborative learning activities to foster interaction. The high positive response for Q4 (Q4: The collaborative and group game-based learning activities encouraged me to better interact with

my peers) in the optional survey among the experimental group (4.8—strongly agree) weighs in on the inference made. Further evidence of a very high positive response (4.9—strongly agree) for Q1 in the optional survey conducted among the experimental group indicated that students in this participant group found the module interesting, resulting in increased classroom attendance and, consequently, in increased engagement with learning. The findings in this study align with similar studies that have reported the effectiveness of game-based learning methods in improving student engagement and student interaction in the classroom environment [85,86]. However, the past studies were pre-dominantly based on digital games, unlike in this study. The findings in this study confirmed that non-digital game-based interventions have a similar impact on student engagement as digital game-based learning, at least among the first-year university student cohorts.

**Table 4.** Findings related to RQ2.

| Data | Experimental Group | Control Group |
| --- | --- | --- |
| Avg weekly attendance for the module (%) | 87% | 74% |
| Avg weekly attendance for three other modules | Module 1 (74%)/Module 2 (70%)/Module 3 (72%) | Module 1 (74%)/Module 2 (70%)/Module 3 (72%) |
| EoM Q3: How actively did you participate in the classroom activities | 4.71 (very actively) | 3.2 (somewhat actively) |
| Survey Q4. The collaborative and group game-based learning activities encouraged me to better interact with my peers | 4.8 (strongly agree) | Data not collected |
| Survey Q1. The game-based learning activities made the module interesting | 4.9 (strongly agree) | Data not collected |

### 6.3. Findings Related to Research Question 3

RQ3: How do in-classroom game-based learning activities impact the overall learning experience of first-year computing students transitioning to higher education?

While academic performance and classroom attendance are fair indicators of learning experience, specific qualitative data were gathered to assess the learning experience of the two participant groups more accurately. The responses for Q2, Q4, and Q5 from the end of module survey (Section 5.4.1) and Q2 from the optional survey questionnaire were in particular considered to assess the learning experience of the cohort (Q2: How difficult (overall) did you find this module?) from the end of the module survey received similar responses from both participant groups. With 1 being the least difficult and 5 the most difficult, the experimental group rated the module 2.3 (moderately difficult), and the control group rated it as 3.1 (difficult). No inference relating to the learning experience could be made with confidence due to the very small variation in the responses to Q2. For further evidence, we refer to the responses to Q4 and Q5 from the end of module questionnaire. Q4 (What did you like about this module?) resulted in 9% of students saying "nothing" within the experimental group compared to 37% within the control group, indicating a much larger number of the cohort were unhappy with the module in the control group than the experimental group. This cannot be directly attributed to game-based learning activities; however, the top three keywords found in the qualitative response for this question within the experimental group were "game-based learning activities", "packet tracer", and "team activities", indicating that the game-based intervention had made an impact on the overall learning experience of the students in the experimental group. Among the control group participants, Q5 (What suggestions can you offer that would help

make your learning experience better in this module?) predominantly resulted in students wanting more hands-on/practical activities in the module, and some students mentioned that the assignment was not interesting. As discussed in Section 2, the review of the end of module survey responses of the previous two presentations of this module was one of the motivating factors for this study and for introducing game-based learning activities in the classroom for the experimental group. Packet Tracer featured in both groups of participants as a contributor for why students liked the module, indicating students enjoyed hands-on activities in the module more than the lectures. While both groups indicated a dislike for the assignment, there was no clear evidence on the reason for the same. This indicates that the experimental group clearly had an enhanced learning experience in the module compared to the control group owing to the hands-on experience and team interaction offered by the game-based learning activities.

### 6.4. Enabling Factors for Successful Transition to Higher Education

In order to discuss the relationship between the findings from this study and factors enabling the successful transition of students in first-year computing in higher education, we considered Tinto's theory of academic integration [5]. The constructs of social and academic integration are an integral part of Tinto's framework to enable student integration into higher education. The concept of academic integration comprises several dimensions that can be broadly identified by academic self-efficacy, academic engagement, peer learning community, and academic commitment. Each of the four metrics measured as a part of this study, i.e., academic performance, learner engagement, classroom interaction, and overall learner experience, has a direct link to the characteristics that result in the academic integration of the student, as indicated in Figure 4 The findings relating to RQ1 indicated a marked spike in academic performance among the experimental participants, and there was sufficient evidence to indicate that the in-classroom game-based interventions were an influencing factor. Past studies indicated that there is a moderate correlation between academic self-efficacy and academic performance [87]. Learning about computer networks and security taught in this module occurs when students are able to form their own concepts enabled by the game-based learning activities that were implemented based on the constructivism approach. Studies indicate critical and creative thinking is shaped by the application of the GBL methods [88]. This is because it encourages students to problem solve and self-learn, leading to students' improved self-esteem and self-efficacy, resulting in improved student achievement in the subject area taught. It can be inferred that the in-classroom game-based interventions contributed to the enhanced academic self-efficacy of the experimental group. Findings from RQ2 indicated that the participants from the experimental group displayed better classroom interaction and higher classroom attendance. This can be attributed with moderate confidence to the in-classroom game-based learning activities, which were designed to incorporate collaborative elements, teamwork features, and challenge-based activities to enhance classroom interaction and learner engagement. Q5 from the optional survey questionnaire that the experimental group answered (Q5: The classroom interaction fostered by the game-based learning activities extended to out-of-class engagement with peers) received a moderately positive response of 3.8 (agree) from the survey participants, which indicated that the game-based interventions somewhat influenced the formation of a peer learning community within the classroom that extended beyond the classroom. Findings from RQ3 although indicated that the in-classroom game-based learning activities contributed to the overall enhanced learning experience of the learners in the experimental group; the study had some limitations to make this inference with much confidence. Previous research has shown that first-year university students are more likely to be successful and transition smoothly into higher education if they develop factors favourable to success including academic self-efficacy, academic engagement, and forming a peer learning community [5,7]. Based on such findings from the previous study, it can be deduced that the in-classroom game-based interventions among the first-year university students could help them successfully

transition into the university learning environment. Figure 4 illustrates this deduction and relationship between the findings from the previous studies and the findings from this study.

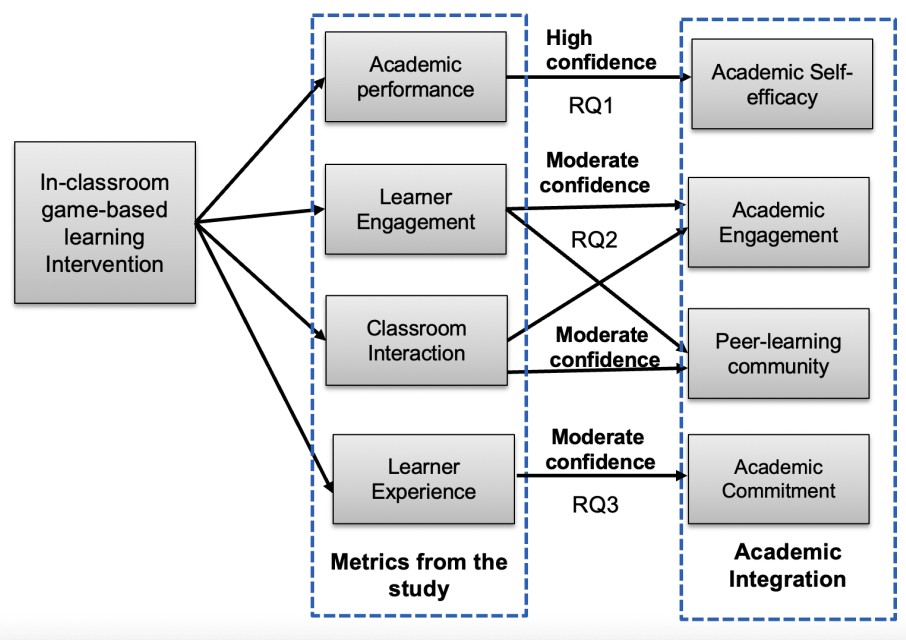

**Figure 4.** Relationship between successful transition to university and findings from this study.

## 7. Limitations of the Study

This study and the findings are specific to the first-year computing students studying the introductory concepts of networking and security at Edge Hill University in England. The study was conducted in 2018; however, the data considered in this study (academic scores, academic attendance, end of module survey, etc.) were gathered between 2016 and 2018. While the data being dated is a limitation, the relevance of the study and some of the findings are valid for current date. The limitation is in the planning of the study, which was originally intended to explore the effectiveness of non-digital game-based learning methods in classroom settings. The original research design did not incorporate factors that contributed to the successful transition of first-year students. This would have allowed the researchers to measure specific metrics and characteristics that would help infer with more confidence the enabling factors to a successful transition. The study lacked sound qualitative data gathering to support the quantitative evidence and to make sound inferences. The study also did not consider any gender or age variance, which may have led to different results. The study participants in this research consisted of all of the first-year university students who had chosen to study computing and a good number of data points for the entire cohort, so the outcomes are likely to be representative of the cohort. The study would have had more impact if the control group and the experimental group were studying in the same academic year and the changes were measured across the entire academic year, instead of within the 12 weeks of the module. The findings from the study relied predominantly on the quantitative data, which are useful for describing the change at a given specific point in time for the specific participant group; however, they did not help understand the "why" of the research study. In a future study, the research design will need to be reviewed to include substantial qualitative data to support the quantitative data analysis. There are likely to be other elements of the students' experience that were not accounted for in this study. Furthermore, the study did not consider any demography factor such as age, gender, and background that may have influenced the findings and the inferences made.

## 8. Conclusions

This study aimed to highlight the need to review the teaching and learning practices within first-year university classroom settings in order to retain Gen Z students that are entering universities and help them successfully transition to higher education. In this context, the study explored the impact of in-classroom non-digital game-based learning techniques on the academic performance, learner engagement, peer interaction in classrooms, and overall learner experience. Taking into account the limitations of this study as discussed in the previous section, the findings from this study indicated with high confidence that in-classroom game-based learning activities are a major contributing factor towards the enhanced academic performance and increased classroom attendance observed within the experimental group, indicating improved academic self-efficacy among the students. While there was some evidence of improved classroom interaction, peer learning, and enhanced engagement among the participants in the experimental group, these can be attributed to the game-based interventions only with moderate confidence due to the lack of sufficient qualitative data and the limitations of the study already discussed. Similarly, the findings from the study indicated an overall enhanced learning experience that enabled the formation of a peer learning community outside the classroom setting; however, they cannot be attributed to the game-based interventions with high confidence.

Further study is needed to infer with confidence the impact of in-classroom game-based interventions on the successful transition of first-year university students, as well as further exploration is required with other groups of students and disciplines. What is particularly relevant in this study was the use of non-digital game-based learning activities within a computing degree qualification. Most information technology and computing qualifications naturally lend themselves to the use of technology-enabled teaching practices. This is reflected in the majority of past studies where game-based learning has been implemented in computing, and allied courses such as programming, computer networking, or cyber security have used digital game-based learning resources. This study indicated that non-digital game-based interventions in the classroom have a similar impact on student engagement, classroom interaction, and academic performance. It may be interred that non-digital game-based interventions have a strong role to play in transforming the learning environment in the classroom from a teacher-centric one to a more student-centric, interactive, and creative learning environment. The positive outcomes reported in this study may be attributed to this change in the classroom learning environment to suit the needs of Gen Z learners. The implementation of non-digital game-based interventions can give students a whole new sense of learning experience compared to chalk and talk. Therefore, the application of game-based learning in teaching and learning of computer networking and security courses should be applied to help improve the quality of teaching and, thus, improve student performance.

Finally, the study makes a significant contribution to pedagogical research relating to the use of non-digital in-classroom game-based learning activities and their potential impact. It contributes towards improved understanding of how the in-classroom learner interaction may potentially extend beyond the classroom environment and could lead to social and academic integration among the first-year university students. The study plays a pivotal role in sharing best practices and introducing new classroom teaching methods incorporating game-based learning activities in other modules taught in the first year of the university to engage young learners and help them transition to higher education. Given the continued induction of a young demography of learners in the first year of university, the significance of game-based learning may open the door to creativity in teaching and learning practices and may significantly help break the barriers to learning among young learners.

**Funding:** This research received no external funding.

**Informed Consent Statement:** Informed consent was obtained from all students involved in the study.

**Data Availability Statement:** The following data is not available due to privacy and ethical restrictions: academic performance data, student attendance data and end of module evaluation data. The data captured from the voluntary survey completed by the experimental group is accessible from—https://docs.google.com/spreadsheets/d/1sjoSM6ALs99Gy2mXc4oyupYWnLPc30iGfRwT0fm-EJQ/edit?resourcekey#gid=1908702840 (accessed on 1 January 2023).

**Acknowledgments:** I would like to acknowledge the support provided by the teaching assistants and students at Edge Hill University, where this study was conducted.

**Conflicts of Interest:** The author declares no conflict of interest.

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
