# Peer review of "The Impact of In-Classroom Non-Digital Game-Based Learning Activities on Students Transitioning to Higher Education"

_education, doi:10.3390/educsci13040328_

Round 1

Reviewer 1 Report

This paper is clearly written but I am not sure of the value of the contribution to the field. The data are quite old (collected during 2016-18) and the literature review only grounds the work in the very basics from the field of game-based learning (e.g. work of Gee, Squire, Prensky - which while innovative at the time is quite dated and does not represent the maturity of the field. And the only reference to general learning theory is constructivism. So the grounding is very thin. 

In addition the first research questions about comparing learning outcomes between a game-based experience and a control appears flawed. If I understand correctly - different outcome measures were used for the different groups (one being game-based and one not) so how can these be compared for results? No validation of these instruments is discussed. The other data relies on student self-report of self-efficacy which is questionable. Finally the most interesting research question is about the impact on transition to higher ed, but there seems to be little framing and no methodology about this aspect of the study. Therefore, it feels like basically a thin replication of the many game-based learning studies that have preceded it.

I would suggest a much deeper dive into the GBL literature of the past decade or so to frame more interesting questions and to find a way to have more comparablecomparablecomparablecomparablecomparablecomparablecomparable measures of the outcomes, focusing more on the impact of transition to higher ed.

Author Response

All comments taken onboard. Please see the attachment for the detailed response. 

Reviewer 2 Report

This manuscript is potentially publishable, includes a big sample and obtains relevant results. However, the authors should review the analysis and the theoretical framework.

Abstract

The summary is adequate.

Introduction

In this study, the authors aim to identify the potential offered by GBL to promote the integration of students into the university and improve their academic performance. The theoretical framework addresses the importance of student integration in the university to avoid dropout. However, it does not indicate what teaching conditions promoted with GBL are useful in learning. They focus on student motivation, but we must consider other teaching conditions that affect learning (the role of the teacher as a guide, an active student body, cooperation, promoting the production and not the reproduction of information, etc.). This teaching, focusing on student-centred learning, should be aimed at promoting competencies.

In the section Motivation for study, it is emphasized that the teaching initially promoted in the Digital Word module did not favour the students' interest in learning. The type of teaching carried out during this period seems to be content-based learning. I think the authors should develop the specific conditions of this type of teaching to conclude by pointing out the importance of promoting student-centred learning. In this section, the authors can include the teaching conditions that should be implemented. In addition, I think it is important to point out the academic year in which the activity described in this section is carried out and the year in which the quasi-experimental study is implemented. I have observed that these data are included in the limitations, but I think they should be incorporated beforehand, at least in the method section.

It is also important to note that millennials are not the most representative population of this study. In 2018, the youngest millennial, born in 1996, would be 22. Therefore, these students would already be finishing their studies. I think it is more advisable to talk about generation Z or highlight the moment of transition in which the sample is obtained.

Method and results

It is no need to include the paragraph on positivism (lines 351-361). On the other hand, in the following lines, the authors point out the use of a mixed research approach. However, although we talk about quantitative methodology, nothing is said about qualitative. How are qualitative results evaluated? Through open-ended questions? How were the qualitative data analyzed? I would recommend using a category system that identifies the most common topics. 

On the other hand, who designs the End of module questionnaire? I think the authors should include this data. As for the optional survey, were any inter-judge analyses performed? Also, authors should consider using the omega index to identify the reliability instead of Crombach's Alpha.

It is also necessary to use a statistic to report whether the differences between the experimental and the control group were significant (e.g., T Student). On the other hand, I recommend including the size of the effect obtained. This aspect should be included in the method and incorporated into the results section. 

Conclusions

The author should include the research limitations in the conclusions section. On the other hand, the conclusions section seems to me to be incomplete. It would be relevant to discuss the results obtained according to what is identified by the scientific research.

References

The scientific literature is a bit obsoleted. It is necessary to include more references that have been published in the last three years. Besides, some references have mistakes.

Author Response

All comments taken on board. Please see the attachment for a detailed response.

Reviewer 3 Report

The article analyses a significant phenomenon in the educational practice. A very specific topic is formulated.

The essential observation is that the part of the research philosophy needs to be revised. It is stated that the research is based on the positivist philosophy, but it is clear that it represents only a quantitative study. It is further written that a mixed research strategy was chosen, but it remains unclear what philosophical direction it is based on. It is indicated that a qualitative study was applied (line 369), but its results cannot be found in the article.

Basically, an experimental research strategy was applied here, and this is the strategy that should be justified. This place in the article needs to be described more clearly and revised.

It would make sense to prepare a part of the discussion in which to present the limitations of the study.

In a general sense, the article is interesting and enriches the understanding with new practices and ideas.

Author Response

(The authors gave the same response as above.)

Reviewer 4 Report

- what is the motivation and support for conducting an in-classroom study? 

- what are the similarities and differences between GBL and PBL? suggest adding and discussing references, such as: Jääskä, E., & Aaltonen, K. (2022). Teachers’ experiences of using game-based learning methods in project management higher education. Project Leadership and Society, 3, 100041.

- line 82: suggest adding and discussing applications of the GenZ and GenY ways of interacting with computers.

- line 197: is it necessary to add an RQ on possible solutions? 

- line 286: which particular game format did the authors focus on? 

- line 350: suggest adding and discussing relevant methods, such as: Lewrick, M., Link, P., & Leifer, L. (2018). The design thinking playbook: Mindful digital transformation of teams, products, services, businesses and ecosystems. John Wiley & Sons.

- line 372: what were the selection criteria and process? 

- line 569: how can other researchers re-use/apply the results in their studies, contexts, and cultures? 

Author Response

(The authors gave the same response as above.)

Round 2

Reviewer 1 Report

The authors have added a few more references about higher education (missing a full body of literature out there) and still argues that there is a gap in the literature in their responses. I simply do not agree. A quick search on google scholar with the keywords "game-based learning" and "higher education" yeilds 182,000 results with many on the first few pages seeming relevant to this work. The theoretical framing of this paper thus still remains weak in my opinion.

Author Response

Please find my response that explains the rationale behind the literature review presented in this paper, the gap in literature that this paper addresses. Hopefully this will address the reviewer’s comment.

As rightly identified by the reviewer, there are several research articles on game-based learning (182000)  published during the past decades. However, not all the literature is relevant to the context of this study which has three specific variants - i) higher education learners  ii) studying computing subject and most importantly the game-based interventions are iii) non-digital game based intervention within classroom setting, unlike many of the interventions that are based on the VLE. It may be noted that the majority of the published work in game-base learning is based on digital game-based learning methods within the context of higher education. In this paper we begin by reviewing the seminal literature on the use of game-based methods within learning context and then narrow the discussion to those work conducted in the context of higher-education and within the discipline of computing or related studies. Then the literature further narrows down to discussing the effectiveness of GBL methods in influencing the student motivation, engagement and that which facilitates social interaction among learners which is relevant to the research question addressed in this paper. We further review the very few literature that is available on non-digital game-based learning methods and illustrate the relative strengths and weaknesses of these methods.   A  total of 50 academic papers have been considered in the literature review within the ’Pedagogical motivator’ section and the ‘literature review’ section. The theoretical framing of this research study in underpinned by this reasonably extensive literature review. It may be noted that this paper contributes to the limited literature available within the context of non-digital game-based learning methods employed among students studying computing or related subject in higher education classroom setting.

Reviewer 2 Report

The authors have made the requested changes to the manuscript. I recommend the acceptance of the manuscript

Author Response

There are no specific amendment requested by the reviewer. All requested amendments were made in the previous submission of the manuscript.